# Team formation amidst conflicts

## ABSTRACT

In this work, we formulate the problem of *team formation amidst conflicts*. The goal is to assign individuals to tasks, with given capacities, taking into account individuals' task preferences and the conflicts between them. Using dependent rounding schemes as our main toolbox, we provide efficient approximation algorithms. Our framework is extremely versatile and can model many different real-world scenarios as they arise in educational settings and human-resource management. We test and deploy our algorithms on real-world datasets and we show that our algorithms find assignments that are better than those found by natural baselines. In the educational setting we also show how our assignments are far better than those done manually by human experts. In the human-resource management application we show how our assignments increase the diversity of teams. Finally, using a synthetic dataset we demonstrate that our algorithms scale very well in practice.

## CCS CONCEPTS

• **Theory of computation** → *Social networks*.

## KEYWORDS

team formation, conflicts, task assignment, diversity

**ACM Reference Format:**
Anonymous Author(s). 2018. Team formation amidst conflicts. In *Proceedings of ACM Web Conference 2024 (WWW '24), (WWW '24)*. ACM, New York, NY, USA, 15 pages. https://doi.org/XXXXXXX.XXXXXXX

## 1 INTRODUCTION

In large project-based classes instructors often need to create *teams* of students and assign them to a finite number of projects they have available. Students are happy if they are in a team with friends and they work on a project they like. Additionally, student teams are efficient if there are no time conflicts between the team members; i.e., conflicts that stem from their class schedule. Traditionally, such assignments are done in an adhoc manner or manually by some admin who can spend several days on the task.

Motivated by such applications in the education domain, we formally define the above problem as a combinatorial optimization problem. For this, we assume two inputs: the *preference graph* and the *conflict graph*. The former captures the preferences of users to projects; this is a bipartite graph with edge weights that are proportional to how much a student likes a project. The latter captures the conflicts between students, i.e., there is a (weighted)

edge between two students if they are incompatible. Our goal is to find an assignment of students to projects such that every student is assigned to one project and every project is not assigned more students than its capacity. The objective is to maximize sum of the weights of the edges of the preference graph that participate in the assignment and and the sum of the weights of the conflict edges across the formed teams. Students assigned to the same project form a team so we call this problem the TEAM FORMATION AMIDST CONFLICTS (TFC) problem and we show it is NP-hard.

In this paper, we present an algorithmic framework for approximating TFC. This framework, consists of two steps: first, our objective is replaced by a concave objective – which can be optimized in polynomial time and produce a fractional solution. Then, the fractional solution is rounded, using dependent-rounding techniques [1, 9]. For our framework to work we need the original objective and its concave relaxation to match on integral inputs.

This general framework is not new. In fact, it is inspired by the *Max-k-Cut with given part sizes* problem [1]. In fact, our problem is identical to the *Max-k-Cut with given part sizes*, except for the fact that we also have an additional linear term in our objective. We show that their approximation algorithm can be applied to our problem. Our contribution is a much more efficient randomized algorithm with better approximation ratio on expectation.

To the best of our knowledge the dependent-rounding techniques we use here, such as pipage and randomized pipage, have not been widely used in practical applications; they primarily stem from work in theoretical computer science [1, 7]. We see the deployment of these techniques in practice as a contribution by itself.

Using real, anonymized data, from large classes [1], we demonstrate that our algorithms work extremely well in practice. In our experiments, we show that the solutions we obtain are much better compared to the manual solutions produced by a course admin across different dimensions and metrics.

Our problem formulation is general and goes beyond educational settings. For example, we can use our framework in human-resource management in order to increase the diversity of departments in companies; depending on the dimension across which we want to diversify we can appropriately define the conflict graph. In our experiments, we show how to achieve gender diversity in a company's departments using this idea.

The generality of our framework calls for efficient algorithms. Part of our contribution is a set of speedup techniques that allow us to apply our approximation algorithms to reasonably large data. In our experimental evaluation we demonstrate that these techniques work extremely well in practice.

**Discussion:** We note here that it was a design decision from our part to consider the conflict graph and maximize conflicts across teams (instead of friends within teams). We believe that this choice gives us greater modeling flexibility to apply our model to a variety of settings. For example, the conflict graph better models time conflicts among collaborators as well as diversity constraints.

---

[1]We obtained an IRB exemption to use the anonymized version of this data.

## 2  RELATED WORK

To the best of our knowledge, we are the first to define and approximate the TFC problem. However, our work is related to works in *team formation*, in the data-mining literature, as well as other *assignment* and *clustering* problems, in the theoretical computer science literature. We review these works below.

**Team formation:** In terms of application, our work belongs to the team-formation literature [2, 3, 12, 16, 17, 20]. Most of these works consider the problem of assigning groups of individuals to tasks (one group per task) such that the tasks are completed and some objective (usually related to the well-functioning of the team or the well-being of the individuals) is optimized. Our problem is a partitioning problem and as such is much more complicated than problems of finding a good team for each available task. Additionally, in many of the existing works the objective function has a well-defined structure (e.g., is monotone and submodular, concave). Our objective does not have such a structure and while this gives us modeling power, optimizing it requires more advanced techniques.

**Partitioning problems:** The *Max-k-Cut with given part sizes* problem [1] served as an inspiration for our model. In fact, our 1/2-approximation algorithm is a very close variant of the algorithm presented there. However, our objective function is slightly different from the one defined by Ageev et al. – due to an additional linear term. This allows us to design algorithms tailored to our problem, which achieve better approximation ratios under certain assumptions. Additionally, we focused on developing scalable algorithms as the running time of the algorithm proposed by Ageev et al. was not a computationally feasible approach.

**Clustering problems:** One can view our problem as a clustering problem with capacity constraints [19, 22]. Our model, however, is quite different from these works both in the objective function and the constraints.

**Assignment problems:** Our problem can be viewed as a generalization of the *weighted assignment* problem [15], where the goal is to assign individuals to tasks taking into account the task preferences of individuals. In our problem, apart from task preferences, we also have a graph capturing the relationships (or conflicts) between individuals. This additional structure increases the complexity of the problem significantly. Our problem is also related to the famous *stable marriage* [18] problem and its variants [8, 11]. However, we don't look for a stable matching. Instead, our goal is to optimize an objective function capturing the overall satisfaction of individuals.

**The metric-labeling problem**: A minimization version of our problem is the *metric-labeling* problem [14], where the goal is to assign one of $k$ labels to each node (i.e., partition the nodes). Every assignment incurs assignment costs (based on the choice of label for each node) and separation costs (based on the choice of labels for "related" nodes). In the capacitated version of metric-labeling [4] we are also given a capacity for each partition. The main disadvantage of the algorithm developed for this version [4] is that the capacity constraints are violated by a multiplicative factor. Also, the algorithm only works for labels with uniform capacities. Finally, the approximation factor of the proposed algorithm depends on the number of labels (i.e. tasks). Defining TFC as a maximization problem allows us to overcome all of the above disadvantages.

## 3  PROBLEM DEFINITION

In this section, we provide the necessary notation and we formally define the problem we solve in this paper.

**Notation:** Throughout the paper, we assume that we are given a weighted (undirected) graph $G = (V, E_G, w)$ with $w : E_G \to \mathbb{R}_{\geq 0}$. More specifically, each node $v \in V$ corresponds to an individual; the weight $w_{uv}$ of an edge $(u, v) \in E_G$ captures the degree of conflict between individuals $u$ and $v$. We call graph $G$ the *conflict graph*.

In addition to the conflict graph $G$, we also assume a preference graph $R$, which is a *bipartite graph*, i.e., $R = (V, T, E_R, c)$. The one side of the graph corresponds to individuals $(V)$, the other side to items or tasks $T$. The edges $(E_R)$ capture the preferences of individuals to projects. More specifically $c : V \times T \to \mathbb{R}_{\geq 0}$ is a *preference* function, where $c_{vt}$ captures the satisfaction of individual $v \in V$ when assigned to task $t \in T$. Without loss of generality we assume that $0 \leq c_{ut} \leq 1$.

Throughout, we assume that each individual $v \in V$ is assigned to exactly one task and that each task $t \in T$ has *capacity* $p_t$, which is task-specific.

**The Team Formation amidst Conflicts problem:** Given the above, our goal is to assign individuals to tasks such that the overall satisfaction of individuals is maximized; the satisfaction of each individual is measured by how much they like the task they are assigned to and the lack of conflicts with the other individuals assigned to the same task. We capture this intuition formally in the form of a (quadratic) program. For this, we define binary variables $x_{vt}$ such that $x_{vt} = 1$ if individual $v$ is assigned to task $t$ and $x_{vt} = 0$ otherwise. Thus, our goal is the following:

$$\max \quad F(\mathbf{x}) = \lambda \sum_{v \in V} \sum_{t \in T} c_{vt} x_{vt} + \sum_{(u,v) \in E_G} w_{uv}\left(1 - \sum_{t \in T} x_{ut} x_{vt}\right) \tag{1}$$

$$\text{s.t.} \quad \sum_{t \in T} x_{vt} = 1, v \in V \tag{2}$$

$$\sum_{v \in V} x_{vt} \leq p_t, t \in T \tag{3}$$

$$x_{vt} \in \{0, 1\}, v \in V, t \in T \tag{4}$$

We call the problem captured by the above program Team Formation amidst Conflicts or TFC for short. The linear term of the objective captures the satisfaction of assigning individuals to tasks and we call it the *task satisfaction term*: $F_R = \sum_{v \in V} \sum_{t \in T} c_{vt} x_{vt}$. The quadratic term captures conflicts in the following sense. The objective increases by $w_{uv}$ whenever there is conflict between individuals $u$ and $v$ and they are assigned to different tasks. We call this term the *social satisfaction term*, i.e., $F_G = \sum_{(u,v) \in E_G} w_{uv}(1 - \sum_{t \in T} x_{ut} x_{vt})$; this term models Max-k-Cut with given sizes of parts [1].

As far as the constraints are concerned: the first constraint enforces that every individual is assigned to exactly one task while the second constraint enforces that we assign at most $p_t$ individuals to task $t \in T$; $p_t$ is the capacity of tasks. Observe that our problem as represented above is a quadratic program with integer constraints and the objective function $F$ is non-convex. This observation hints that the problem may be computationally hard. In fact, we have the following result regarding the hardness of TFC:

LEMMA 1. *The* TEAM FORMATION AMIDST CONFLICTS *problem is NP-hard.*

The proof stems from the fact that our problem includes the MAX-CUT problem [13], which is NP-hard. To see this, consider the instance of our problem that has two tasks $T = \{t_1, t_2\}$ such that $p_{t_1} = p_{t_2} = |V|$. Set $c_{vt} = 0, \forall v \in V, \forall t \in T$. This is indeed an instance of the MAX-CUT problem; this observation concludes the proof of Lemma 1.

An interesting question is what is the value of the hyperparameter $\lambda$ in the linear term and how one should go about setting it. Observe that $\lambda$ balances the relative importance of task preferences and conflicts; when $\lambda = 0$, the linear term vanishes and we only optimize for conflicts. As $\lambda$ grows task preferences become dominant. In general, tuning the hyperparameter is application specific. In Section 4.3, we discuss how we tune $\lambda$ in practice.

## 4 ALGORITHMS

In this section, we provide approximation algorithms for the TFC problem. Our approach for solving this problem is the following: first, we will find a "nice", i.e., *concave*, relaxation of $F$, which we will call $L$. Then, we will optimize $L$ in the fractional domain. That is, we transform the original TFC problem described in Equations (1)-(4) to the following concave program:

$$\max \quad L(\mathbf{x}) \tag{5}$$

$$\text{s.t.} \quad \sum_{t \in T} x_{vt} = 1, v \in V \tag{6}$$

$$\sum_{v \in V} x_{vt} \le p_t, t \in T \tag{7}$$

$$0 \le x_{vt} \le 1, v \in V, t \in T \tag{8}$$

We call this problem RELAXED-TFC and it is clearly very similar to the TFC problem except for two important differences: the objective $F$ is substituted by its relaxation $L$. Also, the integrality constraints are substituted by the corresponding fractional constraints. The relaxations of $F$ we propose have the following two properties, which are key for the results we propose:

PROPERTY 1. $L(\mathbf{x})$ *is concave.*

PROPERTY 2. *For every* $\mathbf{x} \in \{0, 1\}^{|V| \times |T|}$ : $F(\mathbf{x}) = L(\mathbf{x})$. *I.e., the original function and the relaxation agree on the integral values of* $\mathbf{x}$.

Given that $L$ is concave and the constraints are linear, RELAXED-TFC can be solved using *gradient ascent* [5].

Finally, by using appropriate rounding techniques we transform our solution to an integral solution. This general algorithm, which we call Relax-Round is described in Algorithm 1. Relax-Round serves as a template for the approximation algorithms we develop.

We present two algorithms that use two different concave relaxations and rounding schemes, which in turn give different approximation guarantees and come with their own running-time implications.

---

**Algorithm 1** Relax-Round: A general approximation algorithm for the TFC problem.

**Require:** Objective function $F$, Rounding scheme $\Xi$
  Relax: Given $F$ construct a concave fractional relaxation $L$ such that: $F(\mathbf{x}) = L(\mathbf{x})$ for all integral $\mathbf{x}$.
  Optimize: $\mathbf{y}^* = \arg\max_{\mathbf{y}} L(\mathbf{y})$
  Round: $\mathbf{x} = \Xi(\mathbf{y}^*)$

---

### 4.1 A deterministic $\frac{1}{2}$ - approximation algorithm

We start by describing a $\frac{1}{2}$-approximation algorithm for the TFC problem. This algorithm was first introduced by Ageev et al. [1] for the *Max-k-Cut with given part sizes*. However, we present it here for completeness. We call this algorithm Pipage, because it uses *pipage* rounding [1] in order to instantiate Relax-Round. We describe the concave relaxation $L_1$ and pipage rounding below.

**The $L_1$ concave relaxation:**

$$L_1(\mathbf{x}) = \lambda \sum_{v \in V} \sum_{t \in T} c_{vt} x_{vt} + \sum_{(u,v) \in E_G} w_{uv} \min\left(1, \min_t(2 - x_{ut} - x_{vt})\right). \tag{9}$$

We have the following for $L_1$:

PROPOSITION 1 ([1]). $L_1$ *satisfies Properties 1 and 2.*

The proof of Proposition 1 relies on simple algebra (see [1]) and thus omitted.

**Pipage rounding:** pipage rounding takes a fractional solution $\mathbf{y}$ of the TFC problem and transforms it into an integral solution $\mathbf{x}$. The following is a high-level description of the algorithm. For a more thorough analysis and description of the algorithm we refer the reader to Appendix C.1 and the original paper [1].

Pipage rounding is an iterative algorithm; at each iteration the current fractional solution $\mathbf{y}$ is transformed into a new solution $\mathbf{y}'$ with smaller number of non-integral components. Throughout, we will assume that any solution $\mathbf{y}$ is associated with the bipartite graph $H_{\mathbf{y}} = (V, T, E_{\mathbf{y}})$, where the nodes on the one side correspond to individuals, the nodes on the other side to tasks and there is an edge $e(v, t)$ for every pair $(v, t)$ with $v \in V$ and $t \in T$ if and only if $y_{vt} \in (0, 1)$, i.e., $y_{vt}$ is fractional.

Let $\mathbf{y}$ be a current solution of the program and $H_{\mathbf{y}}$ the corresponding bipartite graph. If $H_{\mathbf{y}}$ contains cycles, then set $C$ to be this cycle. Otherwise, set $C$ to be a path whose endpoints have degree 1. Since $H_{\mathbf{y}}$ is bipartite, in both cases $C$ may be uniquely expressed as the union of two matchings $M_1$ and $M_2$. Suppose we increase all components of $\mathbf{y}$ corresponding to edges in $M_1$, while decreasing all components corresponding to edges in $M_2$ until some component reaches an integral value. Denote this solution by $\mathbf{y}_1$. Symmetrically, by decreasing the values of the variables corresponding to $M_1$ and increasing those corresponding to $M_2$, we get solution $\mathbf{y}_2$. We choose the best of these two solutions by calculating $F(\mathbf{y}_1)$ and $F(\mathbf{y}_2)$. We repeat the procedure until all variables are integral.

Observe that each iteration requires evaluating the objective function $F$ twice. This is a significant drawback of this algorithm, especially for large graphs, where the computation of the function is expensive. In the next section we present a randomized version of pipage rounding that overcomes this problem.

**Approximation guarantees:** Following the analysis of Ageev et al. [1] we can show that Pipage is an 1/2-approximation algorithm for the TFC problem. Thus we have:

THEOREM 1 ([1]). *The* Pipage *algorithm is an $\frac{1}{2}$-approximation algorithm for the TFC problem.*

For completeness we present this proof in Appendix A.1

**Running time:** The overall complexity of Pipage consists of the running time of a gradient-ascent algorithm that finds a fractional solution $\mathbf{y}^*$ to the RELAXED-TFC problem plus the running time of pipage rounding. The latter is $O\left((\mathcal{T}_F + |V| + |T|)|E_{\mathbf{y}^*}|\right)$, where $\mathcal{T}_F$ is the time required to evaluate the function $F$ and $E_{\mathbf{y}^*}$ is the number of fractional components of the initial solution $\mathbf{y}^*$. This is because each of the $E_{\mathbf{y}^*}$ steps of pipage rounding requires time $O\left(\mathcal{T}_F + |V| + |T|\right)$ since we run a Depth-First-Search and two evaluations of $F$.

## 4.2 Randomized $\frac{3}{4}$ - approximation algorithm

Here, we present a $\frac{3}{4}$-approximation algorithm for TFC. We call this algorithm RPipage, because we use the randomized pipage rounding in order to instantiate the Relax-Round algorithm.

**The $L_2$ concave relaxation:**

$$L_2(\mathbf{x}) = \lambda \sum_{v \in V} \sum_{t \in T} c_{vt} x_{vt} - w(E_G) + \sum_{(u,v) \in E_G} \sum_{t \in T} w_{uv} \min(1, x_{ut} + x_{vt}).$$

For $L_2$ we have the following:

PROPOSITION 2. *$L_2$ satisfies Properties 1 and 2.*

The proof of Proposition 2 is given in Appendix A.2.

**Randomized pipage rounding** Here, we briefly present the randomized pipage scheme originally proposed by Gandhi [9]. Randomized pipage rounding proceeds in iterations, just like (deterministic) pipage rounding. If $\mathbf{y}$ is the current fractional solution of the rounding algorithm, we calculate $\mathbf{y}_1$ and $\mathbf{y}_2$ (same as in pipage rounding) and then we probabilistically set $y'$ equal to either $\mathbf{y}_1$ or $\mathbf{y}_2$. For more details we refer the reader to Appendix C.2.

**Approximation guarantees:** In order to prove the $\frac{3}{4}$-approximation ratio of RPipage for TFC we need the following Lemma:

LEMMA 2 ([7]). *If we use $\Xi$ to denote the randomized pipage algorithm that rounds a fractional solution $\mathbf{y}$ to an integral solution $\mathbf{x}$, i.e. $\Xi(\mathbf{y}) = \mathbf{x}$, then $\Xi$ satisfies the following properties:*

- $\mathbb{E}_\Xi[\mathbf{x}] = \mathbf{y}$
- $\mathbb{E}_\Xi[(1 - x_{ut})(1 - x_{vt})] \leq (1 - y_{ut})(1 - y_{vt})$, *for all $u, v \in V$ and $t \in T$*

The proof of this lemma is due to Chekuri et al. [7], and thus omitted. The most important consequence of Lemma 2 is the following proposition, the proof of which is given in Appendix A.3

PROPOSITION 3. *Under Assumption 1, for all $\mathbf{x}, \mathbf{y}$ such that $\mathbf{x} = \Xi(\mathbf{y})$ and $\Xi$ being the randomized pipage rounding, we have that:*

$$\mathbb{E}_\Xi[L(\mathbf{x})] \geq \frac{3}{4} L(\mathbf{y})$$

Now, let $\mathbf{y}^*$ be the optimal fractional solution of the RELAXED-TFC problem with objective $L_2$ and $\Xi(\mathbf{y}^*) = \mathbf{x}^*$, with $\Xi$ being the randomized pipage rounding scheme. Also, let $\mathbf{x}_{\text{int}}$ be the optimal solution of the integral problem TFC. Then, it holds that:

$$F(\mathbf{x}^*) = L_2(\mathbf{x}^*) \geq \frac{3}{4} L_2(\mathbf{y}^*) \geq \frac{3}{4} F(\mathbf{x}_{\text{int}}).$$

Thus, we have the following theorem:

THEOREM 2. *Under Assumption 1,* RPipage *is a $\frac{3}{4}$-randomized approximation algorithm for the TFC problem.*

**Running time:** The overall complexity of RPipage consists of the running time of a gradient-ascent algorithm that finds a fractional solution $\mathbf{y}^*$ to the RELAXED-TFC problem with objective $L_2$ plus the running time of the randomized pipage rounding scheme, which is $O((|T| + |V|)|T||V|)$; assuming that the number of tasks $|T| < |V|$, this becomes $O(|T||V|^2)$. In contrast to deterministic pipage rounding, observe that randomized pipage rounding does not require evaluating the objective function. This results in a significant computational speed-up.

**Discussion:** In the future, it would be interesting to examine if *swap rounding* [7], can be used in place of randomized pipage rounding and whether such a scheme can lead to more efficient algorithms. We leave this as an open problem.

## 4.3 Tuning the hyperparameter $\lambda$

In order for Theorem 2 to hold, we need to make the following assumption:

ASSUMPTION 1. *(Balancing Assumption) Consider a feasible fractional solution $\mathbf{y}$. We assume that the following holds:*

$$\lambda \sum_{v \in V} \sum_{t \in T} c_{vt} y_{vt} - w(E_G) \geq 0$$

$$\lambda \geq \frac{w(E_G)}{\sum_{v \in V} \sum_{t \in T} c_{vt} y_{vt}},$$

*where $w(E_G) = \sum_{(u,v) \in E_G} w_{uv}$. If we also assume that $0 \leq c_{vt} \leq 1$ for all $v \in V$ and $t \in T$, then we have*

$$\lambda \geq \frac{w(E_G)}{|V|} = \frac{d_{avg}}{2},$$

*where $d_{avg}$ is the average degree of the nodes in the conflict graph $G$ and we used the fact that $\sum_{t \in T} x_{vt} = 1, \forall v \in V$.*

The above assumption provides a way to tune the balancing parameter $\lambda$. In practice, we do the following: we introduce the *balancing factor* $\alpha \in \mathbb{R}_{>0}$ and we set $\lambda$ to be $\lambda = \alpha \times \frac{d_{avg}}{2}$. In practice, we tune $\alpha$ as follows: for different values of $\alpha$ we evaluate the task and the social satisfaction terms $\left(F_R^{(\alpha)}, F_G^{(\alpha)}\right)$. Then, we pick the value of $\alpha$ that gives the desired balance between the two terms.

## 5 COMPUTATIONAL SPEEDUPS

We discuss here a few methods we use in order to speedup our algorithms. All heuristics we discuss here can be applied to both Pipage as well as RPipage.

**Converting convex to linear programs:** The algorithms we developed in Section 4 are based on the fact that the RELAXED-TFC problem with objective functions $L_1$ and $L_2$ is a concave problem with linear constraints and it can be solved in polynomial time via an application of gradient ascent. In fact, we show that there is a

way to rewrite the Relaxed-TFC problems with objectives $L_1$ and $L_2$ as linear programs, by adding some extra variables.

For the Relaxed-TFC problem with objective $L_1$, this can be done as follows: first, we substitute the term $\min(1, \min_t(2 - x_{ut} + x_{vt}))$ with the new variable $z_{uv}$ and the objective becomes:

$$L_1(\mathbf{x}) = \lambda \sum_{v \in V} \sum_{t \in T} c_{vt} x_{vt} + \sum_{(u,v) \in E_G} w_{uv} z_{uv}.$$

Then we also add the constraints $z_{uv} \leq 1$ and $z_{uv} \leq 2 - x_{ut} - x_{vt}, t \in T$. The full linear program is given in Appendix A.4.

The corresponding linearization of the $L_2$ objective can be done as follows: we substitute the term $\min(1, x_{ut} + x_{vt})$ with a new variable $x_{uvt}$ such that:

$$L_2(\mathbf{x}) = \lambda \sum_{v \in V} \sum_{t \in T} c_{vt} x_{vt} - w(E_G) + \sum_{(u,v) \in E_G} \sum_{t \in T} w_{uv} x_{uvt}.$$

We also add the constraints $x_{uvt} \leq 1$ and $x_{uvt} \leq x_{ut} + x_{vt}$. The complete linear program is given in Appendix A.5.

The advantage of converting the convex problems into linear is that solving a linear program is much more efficient than solving a convex program with linear constraints. In practice, using the Gurobi solver we obtained speedups up to 500x (see Table 4).

**Sparsification:** When the task capacities are small and the conflict graph is dense, a heuristic, we named `Sparsify`, that works well in practice is randomly removing conflict edges. That is, we keep each conflict edge with a certain probability $p$. Otherwise, with probability $(1 - p)$ we discard the edge. This greatly reduces the number of terms we need to evaluate $F_G$ in our objective resulting in computational speedups when optimizing the fractional relaxation.

Note that when $p = 1$, we don't alter the objective. As $p$ decreases, we remove more conflict edges, resulting in computational speedups, although our fractional solution might not be optimal. Selecting $p$ is problem specific. A rule of thumb is that the denser the conflict graph, the lower we can set $p$.

**Compact:** Our intuition, but also our real-world datasets (see Section 6.2 and Appendix D.2), reveal that our data have the following pattern: the complement of the conflict graph, i.e., the friend graph, consists of relatively small densely-connected communities with similar task preferences. Intuitively, for two individuals $u, v$ that belong in the same community and have similar preferences we would expect that the vectors of $x_{ut}$'s and $x_{vt}$'s will be similar for all $t \in T$. Taking this to the extreme: individuals $u, v$ with the same neighbors in the conflict graph $G_G$ and the same preferences for tasks in $T$ should have identical values $x_{ut}, x_{vt}$.

Formally, this is captured in the following theorem, which is proved in Appendix A.6:

**THEOREM 3.** *Consider two individuals $u, v \in V$ which have identical neighbors in $G_G$ (i.e.,$(u, w) \in E_G \Leftrightarrow (v, w) \in E_G$) and have identical project preferences (i.e., $c_{ut} = c_{vt}, \forall t \in T$). Then, there exists an optimal solution $\mathbf{y}$ of Relaxed-TFC such that $x_{ut} = x_{vt}, \forall t \in T$.*

Motivated by the above theorem we define the `Compact` algorithm. On a high-level the idea is to compact densely-connected subgraphs into supernodes. Note that the supernodes need not be nodes that have identical neighborhood in $G$; after all, it may be unreasonable to assume that this will happen in practice. However, using a graph-partitioning algorithm (e.g., spectral clustering [21],

finding dense components [6]) we can partition the original set of nodes into supernodes with similar neighborhoods. Let $S$ be the set of supernodes, which is a partition of the original set of nodes $V$. Then, we create a conflict graph between supernodes; the number of conflict edges between two supernodes $A$ and $B$ is approximately $|A| \times |B|$ (almost every node of $A$ is in conflict with every node of $B$). Thus, we set in this new conflict graph we set the weight of edge $(A, B)$ to be $w_{AB} = |A| \times |B|$ (assuming that each edge of the original graph has unit weight). The next step is to solve the following *compact* Relaxed-TFC problem:

$$\max \quad L(\mathbf{x}) \tag{10}$$

$$\text{s.t.} \quad \sum_{t \in T} x_{vt} = 1, v \in S \tag{11}$$

$$\sum_{v \in S} |v| x_{vt} \leq p_t, t \in T \tag{12}$$

$$0 \leq x_{vt} \leq 1, \forall v \in S, \forall t \in T. \tag{13}$$

where we use $|v|$ to denote the number of simple nodes in the supernode $v$.

Then, we unroll the solution to obtain a fractional solution for the original graph. That is, for each $v \in V$ we set $x_{vt} = x_{St}$, where $S$ is the supernode $v$ belongs to. Finally, we round the fractional solution to obtain an integral solution. Depending on whether we use $L_1$ or $L_2$ as our objective, we then round the fractional solution using `Pipage` or `RPipage` respectively.

## 6 EXPERIMENTS

In this section, we evaluate our framework using both real-world as well as synthetic datasets. The experiments prove the effectiveness and efficiency of our algorithms as well as the versatility of our model to encompass many different real-world scenarios involving assignment problems with conflicts.

### 6.1 Baselines and Setup

**Baselines:** For our experiments we use the following baselines:

`Quadratic`: This is the optimal algorithm, i.e., the algorithm that solves the original TFC problem as expressed in Equations (1)-(4). We use an off-the shelf solver to implement `Quadratic`, but even though this is a powerful solver, we can run `Quadratic` only for small datasets since the solver is asked to optimize a non-convex quadratic function subject to integral constraints.

`Greedy`: The `Greedy` algorithm sequentially assigns an individual to the best team (i.e., the team that maximizes the objective function $F$) given that the constraints are satisfied. The algorithm terminates when all individuals are assigned. We refer the reader to Appendix B for a detailed analysis of `Greedy`.

`Random`: This is an algorithm that randomly assigns each individual to a team until all individuals are assigned.

`Manual`: This is the *manual* assignment of individuals to tasks as made by a human expert, which is available only in some datasets.

**Experimental setup:** Our experimental setup is descried in Appendix D.1. Unless otherwise stated we use the "linearization" speedup presented in Section 5.

**Table 1: Number of nodes $|V|$, tasks $|T|$ and conflict edges $|E|$ for each dataset.**

| Dataset | $|V|$ | $|T|$ | $|E|$ |
|---------|-------|-------|-------|
| *Class-B* | 28 | 7 | 359 |
| *Class-C* | 26 | 6 | 311 |
| *Class-A* | 168 | 14 | 13952 |
| *Class-D* | 37 | 8 | 648 |
| *Company* | 4000 | 4 | 10166 |
| *Synth-TF* | 1000 | 10 | 450454 |

## 6.2 Datasets

For our experiments, we use three different types of data: (a) data of preferences of students with respect to projects and collaborators in educational settings, (b) data from the Bureau of Labor statistics that concern employees and their assignment to company departments and, finally, (c) a synthetic dataset that we use to test the scalability of our algorithms and the speedups described in Section 5. We describe these datasets below. A summary of the characteristics of each dataset is shown in Table 1.

**Education data:** This is data coming from courses in a US institution [2]. In the classes we considered, there were a number of projects (with fixed capacities) available to students and each student filled in a form with their project preferences (each student ranked the projects from best to worst) and their preferences with respect to other students they want to work with in the same project. We have data from four such classes: *Class-A*, *Class-B*, *Class-C* and *Class-D*. These datasets do not contain a conflict graph between students, but instead a *friend* graph (indicative such graphs are shown in Appendix D.2), i.e. two students that have an edge between them are friends and want to work together. We define the conflict graph as the complement of the friend graph. We assign unit weight to each edge of the conflict graph.

Next, we have to construct the preference graph by assigning weight $c_{u,p}$ for each student $u$ and project $t$. Let $rank_u(t) \in [|T|]$ be the rank of project $t$ in student's $u$ preference list (1 is the best, $|T|$ is the worst). We considered the following functions:

- inverse (*Inverse*): $c_{u,t} = \frac{1}{\text{rank}_u(t)}$
- linear-normalized (*LinNorm*): $c_{u,t} = \frac{|T| - \text{rank}_u(t) + 1}{|T|}$

**Employee data:** Based on statistics from the *U.S. Bureau of Labor Statistics* [3] for 2022, we built a dataset of employees in an company. Specifically, we created a company with 4000 employees and four departments: IT, Sales, HR, PR. Each department has 1000 employees. IT and Sales departments are male dominated while HR and PR are female dominated. The distribution of males and females in each department are according to the data in the *Management occupations* section of the *U.S. Bureau of Labor Statistics*. In our experiments, we add conflict edges between all male employees, since our objective is to distribute the male employees more evenly. Equivalently, we could have added conflict edges between all females. Generally, depending on the diversity goal, one can add

---
[2]IRB exception was obtained in order to use an anonymized version of the data.
[3]bls.gov/cps/cpsaat11.htm

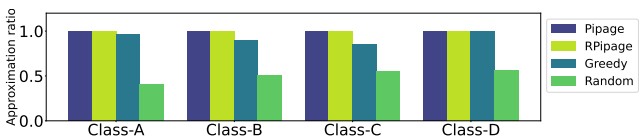

**Figure 1: Education data; approximation ratio of the different algorithms. For all datasets we used the *Inverse* project-preference function and $\alpha = 10$.**

conflicts judiciously to guide the diversification process. For employee preferences, we set $c_{ut} = 1$, if $t$ is the original department of $u$, otherwise $c_{ut} = 0$. We assume that with probability 1% an employee is suitable for switching to a new department. For those employees we set $c_{ut} = 1$, where $t$ is the department to which employee $u$ may switch.

**Synthetic data:** We also created a synthetic dataset (*Synth-TF*) to test the speedups we discussed in Section 5. For this we created $|V| = 1000$ individuals. The conflict graph is defined as the complement of the following friend graph: the friend graph is a planted partition graph where each partition has 100 nodes connected with probability 0.99. Edges across partitions are added with probability $10^{-5}$. For each partition we select a primary project $t$ and set $c_{vt} = 1$ for all nodes $v$ in the partition. Next, for each node $v$ we choose uniformly at random a project $t'$ and set $c_{vt'} = 1$.

## 6.3 Forming teams in education settings

In this section, we evaluate the quantitative and qualitative performance of our algorithms using the education datasets.

**Quantitative performance of our algorithms:** We evaluate the qualitative performance of our algorithms using the *approximation ratio* AR, i.e. the ratio of the objective function evaluated at the solution and the optimal value.

Figure 1 shows the approximation ratios achieved by our algorithms and the different baselines. For all datasets we used the *LinNorm* project preference function. The results for the *Inverse* preference function are presented in Appendix D.3). The results demonstrate that our algorithms have approximation ratio very close to 1.

Interestingly, and despite the fact that in Appendix B we show that the worst-case approximation ratio of Greedy is unbounded, in practice Greedy has AR score very close to 1. We conjecture that this is true due to the correlation between students' friendships and preferences. In fact, we believe that one can bound the approximation ratio of Greedy under such correlation patterns; we leave this as future work.

Somewhat surprisingly, the performance of Random is quite good, although not comparable to the other algorithms. This can be explained by the high density of the conflict graph (or the sparseness of the friend graph) and the fact that the capacities of projects are small relative to the number of individuals. Since the conflict graph is almost a complete graph, small random teams inevitably have large number of conflict edges between them. That said, our analysis demonstrates that the solution given by Random has poor qualitative characteristics.

For these experiments we used $\alpha = 10$ for all datasets, since our primary goal was to assign students to projects they like. Assigning students with their friends was a secondary objective according to the course instructors. We chose the value of $\alpha$ following the grid-search procedure of Section 4.3; the details are given in Appendix D.4. Note that by our selection of $\alpha$, the balancing assumption holds and thus the approximation guarantees of RPipage hold.

**Qualitative performance of our algorithms:** In applications such as the assignment of students to teams, what matters is not just the value of the objective function, but the per-student satisfaction. In this section, we analyze the solutions provided by our algorithms and our baselines and compare those with the *manual* solution provided by a domain expert who tries to find an empirically good assignment based on the input data.

In order to evaluate the quality of the results, we compute the following metrics. Given a solution $\mathbf{x}$ to our problem we define $r(v, \mathbf{x})$ to be the $\text{rank}_v(t)$, where $x_{vt} = 1$, i.e. $r(v, \mathbf{x})$ is the rank – according to $v$ – of the task to which $v$ was assigned. Then, we define the $\mathcal{M}$-preference metric to be:

$$\mathcal{M}Q_R(\mathbf{x}) = \mathcal{M}(r(v, \mathbf{x}) \mid v \in V).$$

In the above equation, $\mathcal{M}$ can be substituted by max or *avg* and the preference metric corresponds to the maximum and average ranking of the projects assigned to students; note that the minimum is not used as it is identical across algorithms and does not provide any insight. Intuitively, the lower the value of $\mathcal{M}Q_R(\mathbf{x}_{\mathcal{A}})$ for a solution provided by an algorithm $\mathcal{A}$, the better the algorithm.

Similarly, we define $\mathcal{M}Q_G(\mathbf{x})$ to be the max or *avg* number of friends (non-conflicts) assigned to students in $v$. In this case, the larger the value $\mathcal{M}Q_G(\mathbf{x}_{\mathcal{A}})$ for a solution provided by an algorithm $\mathcal{A}$, the better the algorithm.

Table 2 shows that students got a better project on average when using the Quadratic, Pipage and RPipage algorithms compared to the Manual assignment. Table 5 shows that students got the same (or almost the same) average number of friends using the Quadratic, Pipage and RPipage algorithms as in the manual assignment.

## 6.4 Forming teams of employees

In this section, we use the *Company* dataset in order to evaluate our algorithms' ability to form diverse teams of employees.

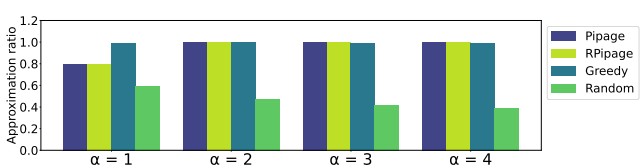

**Figure 2: Employee data; approximation ratio of the different algorithms for $\alpha = 1, 2, 3, 4$.**

**Quantitative performance of our algorithms:** Figure 2 shows the approximation ratios of our algorithms and baselines for the *Company* dataset and for $\alpha = 1, 2, 3, 4$. Somewhat surprisingly, for $\alpha = 1$ our algorithms have a approximation ratio close to 0.8, while Greedy is almost optimal. Random has significantly lower approximation ratio than the other algorithms; most of the times

**Table 2: Education data; $\mathcal{M}Q_R(\mathbf{x}_{\mathcal{A}})$ for $\mathcal{M} = \{max, avg\}$ of assigned project preferences and $\mathcal{A}$ being all algorithms; we also report *std* - the standard deviation for $avgQ_R$; *Inverse* project preference function and $\alpha = 10$.**

| Dataset | Algorithm | $maxQ_R$ | $avgQ_R$ | $std$ |
|---------|-----------|----------|----------|-------|
| Class-A | Quadratic | 14.0 | 1.80 | 2.45 |
|         | Pipage    | 14.0 | 1.82 | 2.51 |
|         | RPipage   | 14.0 | 1.81 | 2.50 |
|         | Greedy    | 14.0 | 1.95 | 2.50 |
|         | Random    | 14.0 | 9.09 | 4.29 |
|         | Manual    | 14.0 | 2.79 | 2.44 |
| Class-B | Quadratic | 4.0 | 1.57 | 0.90 |
|         | Pipage    | 4.0 | 1.57 | 0.90 |
|         | RPipage   | 4.0 | 1.57 | 0.90 |
|         | Greedy    | 7.0 | 2.18 | 1.79 |
|         | Random    | 9.0 | 4.75 | 2.76 |
|         | Manual    | 3.0 | 1.71 | 0.80 |
| Class-C | Quadratic | 6.0 | 1.62 | 1.15 |
|         | Pipage    | 6.0 | 1.62 | 1.15 |
|         | RPipage   | 6.0 | 1.62 | 1.15 |
|         | Greedy    | 6.0 | 2.12 | 1.48 |
|         | Random    | 6.0 | 3.42 | 1.67 |
|         | Manual    | 6.0 | 2.04 | 1.09 |
| Class-D | Quadratic | 2.0 | 1.05 | 0.23 |
|         | Pipage    | 2.0 | 1.06 | 0.23 |
|         | RPipage   | 2.0 | 1.06 | 0.23 |
|         | Greedy    | 2.0 | 1.05 | 0.23 |
|         | Random    | 8.0 | 4.45 | 2.29 |

less than 0.5. In general, all other algorithms have approximation ratio close to 1. It is interesting to observe that in this dataset Greedy performs almost optimally for all choices of $\alpha$; despite the fact that its worst case approximation factor is unbounded (see Appendix B). As we discussed before, we believe that the reason for this is the correlation between conflicts and task preferences.

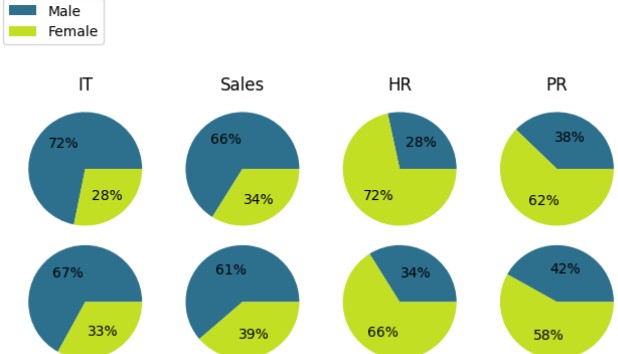

**Figure 3: Employee data; Diversity per department before (1st row) and after (2nd row) we run the Quadratic algorithm ($\alpha = 2$). 8% of the employees changed department. The average male-female percentage gap decreased from 35% to 26%.**

**Table 3: Education data; $MQ_G(x_{\mathcal{A}})$ for $\mathcal{M} = \{max, avg\}$ of number of friends per student and $\mathcal{A}$ being all algorithms; we also report *std* - the standard deviation for $avgQ_G$; *Inverse project preference function and $\alpha = 10$.**

| Dataset | Algorithm | $maxQ_G$ | $avgQ_G$ | $std$ |
|---|---|---|---|---|
| Class-A | Quadratic | 4 | 0.65 | 0.78 |
| | Pipage | 2 | 0.1 | 0.34 |
| | RPipage | 1 | 0.08 | 0.26 |
| | Greedy | 3 | 0.65 | 0.74 |
| | Random | 1 | 0.4 | 0.49 |
| | Manual | 3 | 0.63 | 0.63 |
| Class-B | Quadratic | 2 | 0.64 | 0.77 |
| | Pipage | 2 | 0.64 | 0.77 |
| | RPipage | 2 | 0.57 | 0.78 |
| | Greedy | 3 | 0.79 | 1.08 |
| | Random | 2 | 1.5 | 0.87 |
| | Manual | 2 | 0.64 | 0.67 |
| Class-C | Quadratic | 2 | 0.54 | 0.57 |
| | Pipage | 1 | 0.38 | 0.49 |
| | RPipage | 1 | 0.38 | 0.49 |
| | Greedy | 2 | 0.69 | 0.67 |
| | Random | 1 | 0.54 | 0.5 |
| | Manual | 2 | 0.69 | 0.54 |
| Class-D | Quadratic | 3 | 0.43 | 0.72 |
| | Pipage | 1 | 0.29 | 0.45 |
| | RPipage | 1 | 0.23 | 0.42 |
| | Greedy | 2 | 0.38 | 0.54 |
| | Random | 0 | 0.0 | 0.0 |

**Qualitative performance of our algorithms:** In this dataset, we evaluate the qualitative performance of our framework by showing how the optimal solution to our problem (obtained by Quadratic), affected the diversification of the teams. Figure 3 demonstrates exactly this. While only 8% of the employees changed department, the average male-female percentage gap decreased from 35% to 26%. Varying the value of $\alpha$ we can control this balance. Specifically, if we decrease the value of $\alpha$ the number of people who change department increases, while the average male-female gap decreases. Hyperparameter tuning for the *Company* dataset is further discussed in Appendix D.7.

## 6.5 Evaluating the speedup techniques

In this section, we use the *Synth-TF* dataset to demonstrate the speedups obtained by the different techniques we discussed in Section 5. As we discussed, solving the TFC problem as described in Equations (5)-(8) using a convex solver does not scale up. Thus, we apply to this original problem sparsification and then run the convex solver; we call this algorithm Sparsify-Concave. For the Sparsify algorithm we used $p = 0.01$, i.e., we kept only 1% of the edges of the conflict graph. Alternatively, we transform the problem into a problem with a linear objective by adding auxiliary variables and constraints (as discussed in Section 5) and run the RPipage algorithm. We call this algorithm Linear. Another algorithm we use is Sparsify-Linear, which combines sparsification and linearization. Finally, we also combine Compact with Linear to obtain

**Table 4: Running time (seconds) for the speed-ups**

| Algorithm | Time (seconds) |
|---|---|
| Concave | time out |
| Sparsify-Concave | 1095 |
| Linear | 342 |
| Sparsify-Linear | 2 |
| Compact-Linear | 3 |

the Compact-Linear algorithm. Note that for the implementation of Compact we use the spectral clustering algorithm ([21]) available in scikit-learn [4].

Figure 4 shows the approximation ratios of the above heuristics and Table 4 the running times of the same heuristics on the *Synth-TF* dataset. For this experiment, we set the value of $\alpha = 10$.

Although all algorithms perform almost optimally, speed-ups vary. First, trying to directly optimize the Concave relaxation results in a time-out of our solver. Using Sparsify before optimizing the Concave relaxation renders the problem solvable in 1000 seconds. Using the Linear algorithm yields an extra 3x speedup. Finally, combining Linear with either Sparsify or Compact further reduces the running time down to $2 - 3$ seconds which is a 100x speedup. In total, we managed to reduce the time from 1095 seconds using Sparsify-Concave to $2 - 3$ seconds combining Linear with one of Sparsify or Compact.

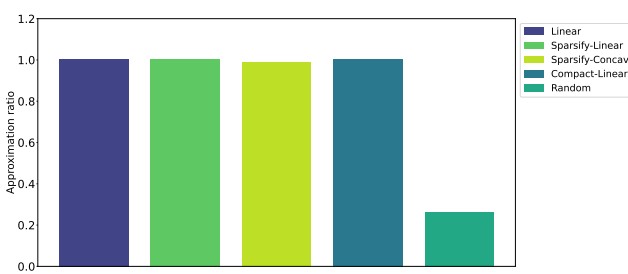

**Figure 4: *Synth-TF* dataset; approximation ratios of the speedups; We used $\alpha = 10$.**

## 7 CONCLUSIONS

Motivated by the need to form teams of students in large project-based classes we defined the TFC problem and showed that (a) it is NP-hard and that (b) it is closely related to Max-$k$-Cut with given part sizes [1]. For TFC, we designed a new efficient randomized approximation algorithm and practical methods for speeding it up. We applied our algorithms to real-world datasets and demonstrated their efficacy across different dimensions.

In the future, we want to further explore possible speedups for our algorithm and also formally investigate the extremely good performance of Greedy in practice – despite its unbounded worst-case approximation ratio.

---

[4]scikit-learn.org/stable/modules/generated/sklearn.cluster.SpectralClustering.html

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

## A  PROOFS

### A.1  Proof of Theorem 1

We begin with the following lemma:

Lemma 3. *Let* $\mathbf{y}$ *be a (fractional) feasible solution to the TFC-1 problem. Then, for every* $u, v \in V$ *we have:*

$$1 - \sum_{t \in T} y_{uv} y_{vt} \geq \frac{1}{2} \min \left( 1, \min_{t \in T} (2 - y_{uv} - y_{vt}) \right)$$

Proof. Let $z_{uv} = \min \left( 1, \min_t (2 - y_{uv} - y_{vt}) \right)$. Define

$$t' = \arg\max_{t \in T} (y_{ut} + y_{vt})$$

and

$$q_{uv} = y_{ut'} + y_{vt'}. \tag{14}$$

Then,

$$z_{uv} = \min(1, 2 - q_{uv}). \tag{15}$$

Also, it holds that

$$\sum_{t \in T} y_{ut} + y_{vt} = \sum_{t \in T} y_{ut} + \sum_{t \in T} y_{vt} = 1 + 1 = 2 \tag{16}$$

as well as the arithmetic-geometric mean inequality which says that for any two positive numbers $a, b$:

$$\left( \frac{a + b}{2} \right)^2 \geq ab. \tag{17}$$

Thus, we have:

$$1 - \sum_{t \in T} y_{ut} y_{vt} = 1 - y_{ut'} y_{vt'} - \sum_{t \neq t'} y_{ut} y_{vt}$$

$$\geq 1 - \left( \frac{y_{ut'} + y_{vt'}}{2} \right)^2 - \sum_{t \neq t'} \left( \frac{y_{ut} + y_{vt}}{2} \right)^2 \quad \text{(using 17)}$$

$$\geq 1 - \left( \frac{q_{uv}}{2} \right)^2 - \left( \frac{\sum_{t \neq t'} y_{ut} + y_{vt}}{2} \right)^2 \quad \text{((14) and convexity)}$$

$$= 1 - \left( \frac{q_{uv}}{2} \right)^2 - \left( \frac{2 - (y_{ut'} + y_{vt'})}{2} \right)^2 \quad \text{(using (16))}$$

$$= 1 - \left( \frac{q_{uv}}{2} \right)^2 - \left( 1 - \frac{q_{uv}}{2} \right)^2 \quad \text{(using (14))}$$

$$= q_{uv} - \frac{q_{uv}^2}{2}.$$

*Case 1.* $1 \leq q_{uv} \leq 2$. Then, by Eq. (15), $z_{uv} = 2 - q_{uv}$, and

$$1 - \sum_{t \in T} y_{ut} y_{vt} \geq q_{uv} - \frac{q_{uv}^2}{2} \geq \frac{1}{2} z_{uv}. \tag{18}$$

*Case 2.* $0 \leq q_{uv} \leq 1$. Then, by Eq. (15), $z_{uv} = 1$. By the assumption of this case, for every $t$ it holds that

$$0 \leq y_{ut} + y_{vt} \leq 1. \tag{19}$$

Using the arithmetic-geometric mean inequality (Eq. (17)), we have

$$1 - \sum_{t \in T} y_{ut} y_{vt} \geq 1 - \sum_{t \in T} \left( \frac{y_{ut} + y_{vt}}{2} \right)^2$$

$$= 1 - \frac{1}{4} \sum_{t \in T} (y_{ut} + y_{vt})^2$$

$$\geq 1 - \frac{1}{4} \sum_{t \in T} (y_{ut} + y_{vt}) \quad \text{(using (16))}$$

$$= 1 - \frac{1}{4} 2$$

$$= \frac{1}{2} z_{uv}$$

$\square$

A corollary of Lemma 3 is the following:

Corollary 1. *If* $\mathbf{y}$ *is a (fractional) feasible solution to problem* Relaxed-TFC *with objective* $L_1$*, then*

$$F(\mathbf{y}) \geq \frac{1}{2} L_1(\mathbf{y}).$$

Let $F^*$ be the value of the optimal (integral) solution to TFC. If $\mathbf{y}^*$ is the optimal (fractional) solution to Relaxed-TFC with objective $L_1$, it holds that $L_1(\mathbf{y}^*) \geq F^*$. From Corollary 1 we have that $F(\mathbf{y}^*) \geq \frac{1}{2}L_1(\mathbf{y}^*)$. Using pipage rounding we can round the fractional solution $\mathbf{y}^*$ to an integral solution $\bar{\mathbf{x}}$ such that $F(\bar{\mathbf{x}}) \geq F(\mathbf{y}^*)$. Thus,

$$F(\bar{\mathbf{x}}) \geq F(\mathbf{y}^*) \geq \frac{1}{2}L_1(\mathbf{y}^*) \geq \frac{1}{2}F^*.$$

## A.2 Proof or Proposition 2

**Property 1** To prove concavity it suffices to see that $L_2(\mathbf{x})$ is the sum of concave functions. Note that $\min(1, x_{ut} + x_{vt})$ is concave.

**Property 2**: For the proof of Property 2, we need the following lemmas:

Lemma 4. *For all* $x, y \in \{0, 1\}$ *it holds that*

$$1 - (1 - x)(1 - y) = x + y - xy = \min(1, x + y).$$

The proof of this lemma consists of checking equality for all combinations of integral values of $x$ and $y$.

Lemma 5 ([10]). *For all* $x, y \in [0, 1]$ *it holds that*

$$1 - (1 - x)(1 - y) \geq \frac{3}{4}\min(1, x + y).$$

Now we are ready to prove the following:

Proposition 4. *For all integral* $\mathbf{x}$ *it holds that* $F(\mathbf{x}) = L_2(\mathbf{x})$.

Proof. It holds that

$$\sum_{(u,v) \in E} w_{uv}(1 - \sum_{t \in T} x_{ut}x_{vt}) =$$

$$w(E) - \sum_{(u,v) \in E}\sum_{t \in T} w_{uv}x_{ut}x_{vt} =$$

$$w(E) + \sum_{(u,v) \in E}\sum_{t \in T} w_{uv}\min(1, x_{ut} + x_{vt})$$

$$- \sum_{(u,v) \in E}\sum_{t \in T} w_{uv}(x_{ut} + x_{vt}) =$$

$$w(E) + \sum_{(u,v) \in E}\sum_{t \in T} w_{uv}\min(1, x_{ut} + x_{vt}) - 2w(E) =$$

$$\sum_{(u,v) \in E}\sum_{t \in T} w_{uv}\min(1, x_{ut} + x_{vt}) - w(E),$$

where in the second equality we used lemma 4.
Using the above we have that

$$F(\mathbf{x}) = \lambda\sum_{v \in V}\sum_{t \in T} c_{vt}x_{vt} + \sum_{(u,v) \in E} w_{uv}(1 - \sum_{t \in T} x_{ut}x_{vt})$$

$$= \sum_{(u,v) \in E}\sum_{t \in T} w_{uv}\min(1, x_{ut} + x_{vt}) + \lambda\sum_{v \in V}\sum_{t \in T} c_{vt}x_{vt} - w(E)$$

$$= L_2(\mathbf{x}).$$

□

## A.3 Proof of Proposition 3

Proof.

$$\mathbb{E}_\Xi[L(\mathbf{x})] = \mathbb{E}_\Xi\Big[\sum_{(u,v) \in E}\sum_{t \in T} w_{uv}\min(1, x_{ut} + x_{vt})$$

$$+ \lambda\sum_{v \in V}\sum_{t \in T} c_{vt}x_{vt} - w(E)\Big]$$

$$= \mathbb{E}_\Xi\Big[\sum_{(u,v) \in E}\sum_{t \in T} w_{uv}(1 - (1 - x_{ut})(1 - x_{vt}))$$

$$+ \lambda\sum_{v \in V}\sum_{t \in T} c_{vt}x_{vt} - w(E)\Big]$$

$$\geq \sum_{(u,v) \in E}\sum_{t \in T} w_{uv}(1 - (1 - y_{ut})(1 - y_{vt}))$$

$$+ \lambda\sum_{v \in V}\sum_{t \in T} c_{vt}y_{vt} - w(E)$$

$$\geq \frac{3}{4}\sum_{(u,v) \in E}\sum_{t \in T} w_{uv}\min(1, y_{ut} + y_{vt})$$

$$+ \frac{3}{4}\Big(\lambda\sum_{v \in V}\sum_{t \in T} c_{vt}y_{vt} - w(E)\Big)$$

$$= \frac{3}{4}L(\mathbf{y})$$

where in the second line we used Lemma 4, in the third line we used the properties of the rounding operator, and in the fourth line we used Lemma 5 and Assumption 1. □

## A.4 Linearization of $L_1$

$$\max \quad L_1(\mathbf{x}) = \lambda\sum_{v \in V}\sum_{t \in T} c_{vt}x_{vt} + \sum_{(u,v) \in E_G} w_{uv}z_{uv} \quad (20)$$

$$\text{s.t.} \quad \sum_{t \in T} x_{vt} = 1, \text{ for every } v \in V \quad (21)$$

$$\sum_{v \in V} x_{vt} \leq p_t, \text{ for every } t \in T \quad (22)$$

$$z_{uv} \leq 1, \text{ for every } (u, v) \in E_G, \quad (23)$$

$$z_{uv} \leq 2 - x_{ut} - x_{vt}, \text{ for every } (u, v) \in E_G, t \in T. \quad (24)$$

## A.5 Linearization of $L_2$

$$\max \quad L_2(\mathbf{x}) = \sum_{(u,v) \in E_G}\sum_{t \in T} w_{uv}x_{uvt} + \lambda\sum_{v \in V}\sum_{t \in T} c_{vt}x_{vt} - w(E_G)$$
$$(25)$$

$$\text{s.t.} \quad \sum_{t \in T} x_{vt} = 1, \text{ for every } v \in V \quad (26)$$

$$\sum_{v \in V} x_{vt} \leq p_t, \text{ for every } t \in T \quad (27)$$

$$x_{uvt} \leq 1, \text{ for every } (u, v) \in E_G, t \in T \quad (28)$$

$$x_{uvt} \leq x_{ut} + x_{vt}, \text{ for every } (u, v) \in E_G, t \in T. \quad (29)$$

## A.6 Proof of Theorem 3

Proof. Let $\mathbf{y}$ be an optimal solution of the fractional relaxation of TFC such that it does not hold that $y_{ut} = y_{vt}, \forall t \in T$. Since $\sum_t y_{ut} = \sum_t y_{vt} = 1$, there exist $t_1, t_2 \in T$ such that $y_{ut_1} \neq y_{vt_1}$

and $y_{ut_2} \neq y_{vt_2}$. Let $\tilde{\mathbf{y}}$ be the "symmetrical" solution where we have replaced $u$ with $v$ and vice versa. That is,

- $\tilde{y}_{ut} = y_{vt}, t \in T$
- $\tilde{y}_{vt} = y_{ut}, t \in T$
- $\tilde{y}_{uv't} = y_{vv't}, t \in T, (u, v') \in E$
- $\tilde{y}_{vv't} = y_{uv't}, t \in T, (v, v') \in E$

For the last two equalities note that since nodes $u$ and $v$ are *symmetrical* it holds that $(u, v') \in E \Leftrightarrow (v, v') \in E$. Then, $\mathbf{x} = (\mathbf{y} + \tilde{\mathbf{y}})/2$ is an optimal feasible solution where $x_{ut} = x_{vt}, \forall t \in T$. To see that note that the constraints are linear and thus any convex combination of feasible solutions is also feasible. Moreover, the objective function is linear and thus any convex combination of optimal solutions is also optimal. Finally, concluding our proof observe that

$$x_{ut_1} = x_{vt_1} = \frac{y_{ut_1} + y_{vt_1}}{2}$$

and

$$x_{ut_2} = x_{vt_2} = \frac{y_{ut_2} + y_{vt_2}}{2}$$

□

## B  ANALYSIS OF THE GREEDY ALGORITHM

**Unbounded approximation ratio of Greedy:** We have the following result in terms of the performance of greedy with respect to our objective function:

PROPOSITION 5. Greedy *has unbounded approximation ratio.*

PROOF. Consider the following instance of our problem. $V = \{u, v, z\}$, with task preferences $c_{ut_1} = 1 - \epsilon$, $c_{vt_2} = \epsilon$ and all other preferences equal to 0. $T = \{t_1, t_2\}$, with capacities $p_{t_1} = 1$ and $p_{t_2} = 2$. The conflict graph consists of the edge $(v, z)$ with weight $w_{vz} = \mathcal{W}$. For the objective function assume that $\lambda = 1$. Running the Greedy yields the assignment: $u$ assigned to $t_1$ and $v, z$ are assigned to $t_2$. The optimal assignment is $z$ assigned to $t_1$ and $u, v$ to $t_2$. The approximation ratio is:

$$\text{AR} = \frac{(1 - \epsilon) + \epsilon}{\mathcal{W} + \epsilon} \leq \frac{1}{\mathcal{W}}.$$

As $\mathcal{W} \to \infty$, the approximation ratio AR $\to 0$. □

**Running time of Greedy:** The complexity of Greedy is $O(|V|^2|T|\mathcal{T}_F)$, where $\mathcal{T}_F$ is the cost of calculating $F$. We have $|V|$ iterations in total, since at each iteration we assign one individual to a team. Each iteration costs $|V||T|\mathcal{T}_F$, since we calculate the change in the objective function when considering the addition of each remaining individual to each team which is not full.

## C  DEPENDENT ROUNDING SCHEMES

### C.1  Pipage rounding

In this section we give a description of the pipage rounding algorithm. For a more detailed analysis we refer the reader to the original paper [1]. Pipage rounding is an iterative algorithm; at each iteration the current fractional solution $\mathbf{y}$ is transformed into a new solution $\mathbf{y}'$ with smaller number of non-integral components. Throughout, we will assume that any solution $\mathbf{y}$ is associated with the bipartite graph $H_{\mathbf{y}} = (V, T, E_{\mathbf{y}})$, where the nodes on the one side correspond to individuals, the nodes on the other side to tasks

and there is an edge $e(v, t)$ for every pair $(v, t)$ with $v \in V$ and $t \in T$ if and only if $y_{vt} \in (0, 1)$, i.e., $y_{vt}$ is fractional.

Let $\mathbf{y}$ be a current solution satisfying the constraints of the program and $H_{\mathbf{y}}$ the corresponding bipartite graph. If $H_{\mathbf{y}}$ contains cycles, then set $C$ to be this cycle. Otherwise, set $C$ to be a path whose endpoints have degree 1. Since $H_{\mathbf{y}}$ is bipartite, in both bases $C$ may be uniquely expressed as the union of two matchings $M_1$ and $M_2$. Given this, define a new solution $\mathbf{y}(\epsilon, C)$ as follows:

- if $e \in E_{\mathbf{y}} \setminus C$, then $y_e(\epsilon, C) = y_e$.
- Otherwise, $y_e(\epsilon, C) = y_e + \epsilon$, $e \in M_1$ and $y_e(\epsilon, C) = y_e - \epsilon$, $e \in M_2$.

For the above, set

$$\epsilon_1 = \min\{\epsilon > 0 : (\exists e \in M_1 : y_e + \epsilon = 1) \vee (\exists e \in M_2 : y_e - \epsilon = 0)\}$$

and

$$\epsilon_2 = \min\{\epsilon > 0 : (\exists e \in M_1 : y_e - \epsilon = 0) \vee (\exists e \in M_2 : y_e + \epsilon = 1)\}.$$

Let $\mathbf{y}_1 = \mathbf{y}(-\epsilon_1, C)$ and $\mathbf{y}_2 = \mathbf{y}(\epsilon_2, C)$. Set $\mathbf{y}' = \mathbf{y}_1$, if $F(\mathbf{y}_1) > F(\mathbf{y}_2)$, and $\mathbf{y}' = \mathbf{y}_2$ otherwise. Note that $\mathbf{y}'$ has smaller number of fractional components than $\mathbf{y}$ and, thus, Pipage terminates after at most $|E_{\mathbf{y}^*}|$ iterations, i.e., as many as the number of fractional values in the $\mathbf{y}^*$ vector output by the optimization algorithm. The following theorem states that $\mathbf{y}'$ satisfies the following constraints:

THEOREM 4 ([1]). *Consider performing pipage roudning starting from the fractional solution* $\mathbf{y}$. *Let* $\mathbf{x}$ *be the integral solution produced when* Pipage *terminates. Then,*

$$\left\lfloor \sum_{e(v,t) \in \delta(v)} y_{vt} \right\rfloor \leq \sum_{e(v,t) \in \delta(v)} x_{vt} \leq \left\lfloor \sum_{e(v,t) \in \delta(v)} y_{vt} \right\rfloor + 1,$$

*where for every* $v \in V$ $\delta(v)$ *is the set of edges in the preference graph* $R$, *that are incident to* $v$.

Since all $p_t$'s are integers (see Eq. 3), the above theorem implies that $\mathbf{x}$ is a feasible solution.

### C.2  Randomized pipage rounding

Here, we briefly present the randomized pipage scheme originally proposed by Gandhi [9] adapted to our problem. Randomized pipage rounding proceeds in iterations, just like (deterministic) pipage rounding. If $\mathbf{y}$ is the fractional solution at the current iteration of the rounding algorithm, we update $\mathbf{y}$ as follows:

If $e \in E_{\mathbf{y}} \setminus C$, then $y_e(\epsilon, C) = y_e$. If $e \in C$, then $\mathbf{y}' = \mathbf{y}_1$, with probability $\epsilon_2/(\epsilon_1 + \epsilon_2)$. Otherwise, with probability $\epsilon_1/(\epsilon_1 + \epsilon_2)$, $\mathbf{y}' = \mathbf{y}_2$. Note $C$, $\epsilon_1$, $\epsilon_2$, $\mathbf{y}_1$ and $\mathbf{y}_2$ are the same as the ones defined in the description of pipage rounding. As the number of fractional elements of $\mathbf{y}$ decrease in every iteration, randomized pipage rounding terminates after at most $O(|E_{\mathbf{y}^*}|)$ iterations, where $\mathbf{y}^*$ is the solution to the RELAXED-TFC problem with objective $L_1$.

## D  EXPERIMENTS

### D.1  Experimental setup

All of the experiments were run on a machine with an Intel(R) Xeon(R) Gold 6242 CPU @ 2.80GHz and 16GB memory. All of our code is written in Python 3.6.8. For linear and quadratic optimization we used Gurobi [5]. For optimizing concave functions we used

---

[5]https://www.gurobi.com/

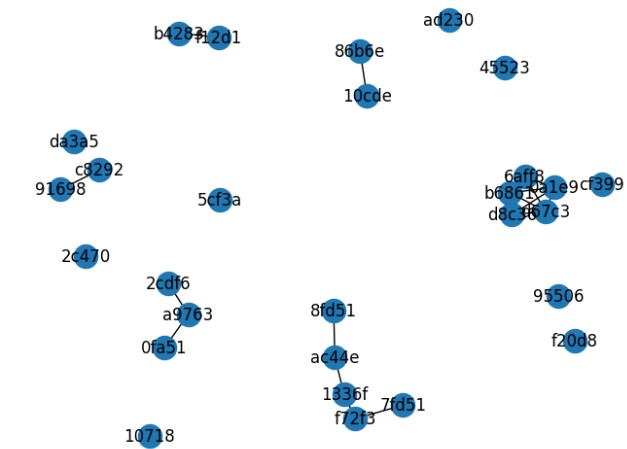

**Figure 5: Friend graph for dataset *Class-B*. On top of nodes are the anonymized student ids.**

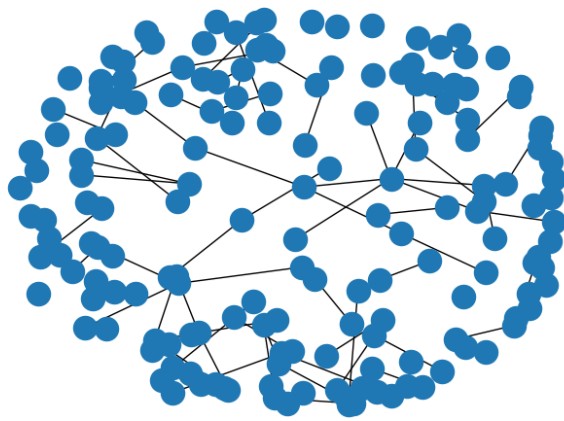

**Figure 6: Friend graph for dataset *Class-A*.**

CVXPY [6]. In all our experiments, unless otherwise explicitly stated, we use the "linearization" speedup we presented in Section 5. Combining this with the Gurobi solver we obtain reasonable running times for all our experiments. We also demonstrate the effect of this linearization in Section 6.5. In fact, when we tried optimizing the concave relaxation with linear constraints as expressed in Equations (9)-(8), our solver could not terminate. For this concave problem we used CVXPY as Gurobi only solves linear (and quadratic) problems. Our code will be made publicly available.

### D.2 Indicative education datasets and their characteristics

Figures 5 and 6 show the friend graphs (i.e., the complement of the conflict graphs) which is the input in the education datasets. Observe the sparsity of both friend graphs.

Figure 7 and figure 8 show the friend graph for *Class-A* and *Class-B* respectively, where the nodes with the same color are assigned to the same project.

### D.3 Approximation ratios for *LinNorm* preference function

Figure 9 shows the approximation ratios for the *LinNorm* preference function.

### D.4 Hyperparameter tuning for the education datasets

In Section 4.3, we gave a preview of how to tune the hyperparameter $\lambda$ or $\alpha$ by computing the terms $\left(F_G^{\alpha}, F_R^{\alpha}\right)$ for different values of $\alpha$. This results in an "elbow" plot that illustrates the trade-off between the two terms and allows us to tune the hyperparameter in a informed way. We present the plots for the *LinNorm* project preference function in Figure 10; the plots for the *Inverse* project preference function are very similar and are presented in Figure 11.

[6]https://www.cvxpy.org/

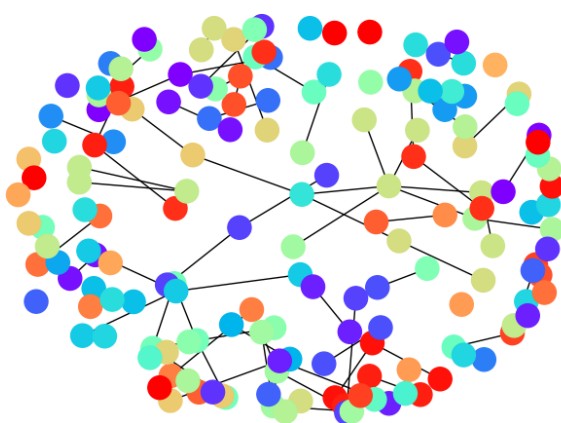

**Figure 7: Friend graph colored using the optimal team assignment (Quadratic) for *Class-A*. Labels are omitted for clarity.**

Note that when $\alpha = 0$, the conflict term is almost equal to the number of conflict edges, i.e. almost all individuals in conflict are placed in different teams. When $\alpha = 10$, the project preference term is almost equal to the number of individuals, i.e. almost all individuals get their highest-ranked project. This is because $c_{vt} = 1$, if the highest-ranked project of $v$ is $t$.

### D.5 Qualitative results for education datasets using the *Inverse* project preference function

Table 5 shows that students got the same (or almost the same) average number of friends using the Quadratic, Pipage and RPipage algorithms as in the Manual assignment.

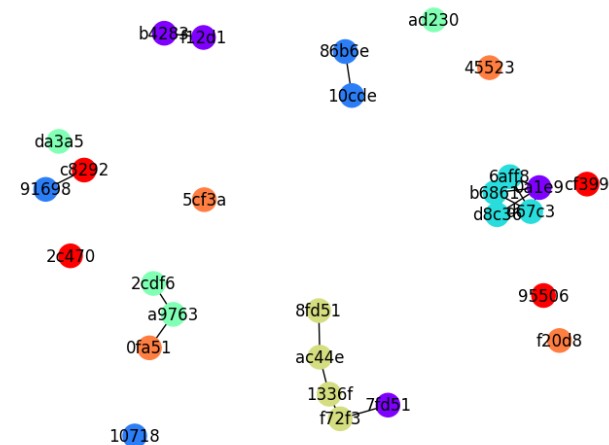

Figure 8: Friend graph colored using the optimal team assignment (Quadratic) for *Class-B*. On top of nodes are the anonymized student ids.

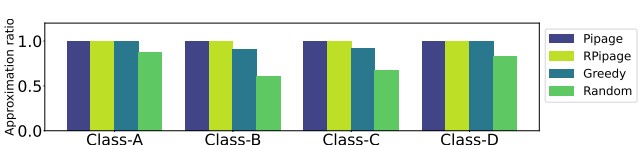

Figure 9: Education data; approximation ratios for education datasets. For all datasets we used the *LinNorm* project preference function and $\alpha = 10$.

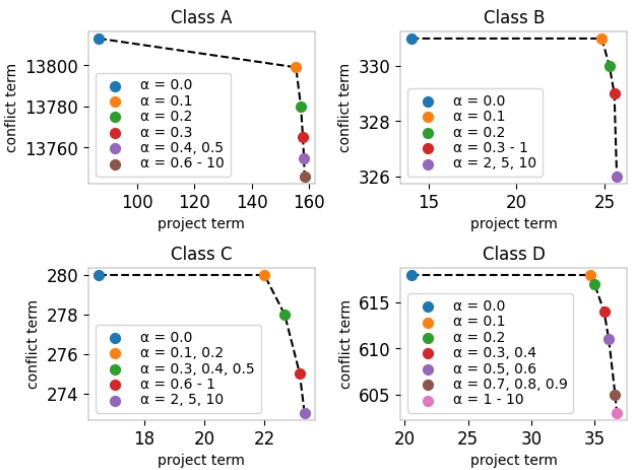

Figure 10: Education data; hyperparameter $\alpha$ tuning. We used the *LinNorm* project preference function.

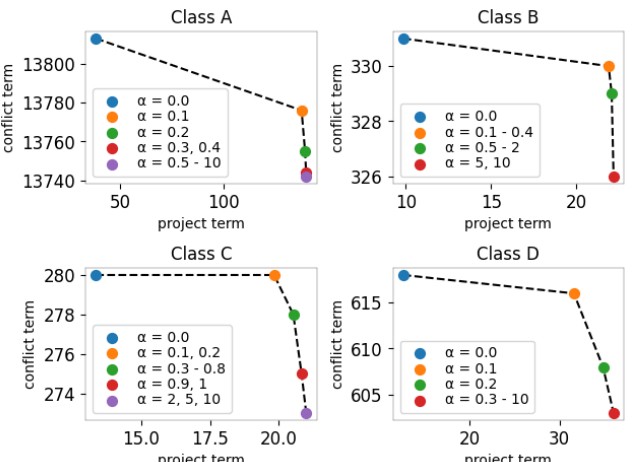

Figure 11: Education data; hyperparameter $\alpha$ tuning. We used the *Inverse* project preference function.

Table 5: Education data; $\mathcal{M}Q_G(\mathbf{x}_{\mathcal{A}})$ for $\mathcal{M} = \{max, avg\}$ of number of friends per student and $\mathcal{A}$ being all algorithms; we also report *std* - the standard deviation for *avgQ*. For all datasets we used the *Inverse* project preference function and $\alpha = 10$.

| Dataset | Algorithm | max$Q_G$ | avg$Q_G$ | std |
|---|---|---|---|---|
| *Class-A* | Quadratic | 4 | 0.65 | 0.78 |
| | Pipage | 2 | 0.1 | 0.34 |
| | RPipage | 1 | 0.08 | 0.26 |
| | Greedy | 3 | 0.65 | 0.74 |
| | Random | 1 | 0.4 | 0.49 |
| | Manual | 3 | 0.63 | 0.63 |
| *Class-B* | Quadratic | 2 | 0.64 | 0.77 |
| | Pipage | 2 | 0.64 | 0.77 |
| | RPipage | 2 | 0.57 | 0.78 |
| | Greedy | 3 | 0.79 | 1.08 |
| | Random | 2 | 1.5 | 0.87 |
| | Manual | 2 | 0.64 | 0.67 |
| *Class-C* | Quadratic | 2 | 0.54 | 0.57 |
| | Pipage | 1 | 0.38 | 0.49 |
| | RPipage | 1 | 0.38 | 0.49 |
| | Greedy | 2 | 0.69 | 0.67 |
| | Random | 1 | 0.54 | 0.5 |
| | Manual | 2 | 0.69 | 0.54 |
| *Class-D* | Quadratic | 3 | 0.43 | 0.72 |
| | Pipage | 1 | 0.29 | 0.45 |
| | RPipage | 1 | 0.23 | 0.42 |
| | Greedy | 2 | 0.38 | 0.54 |
| | Random | 0 | 0.0 | 0.0 |

## D.6 Qualitative results for education datasets using the *LinNorm* project preference function

Tables 6 and 7 contain the qualitative results for the *LinNorm* project preference function. Our algorithms are again comparable or better

**Table 7: Education data; $\mathcal{M}Q_G(x_{\mathcal{A}})$ for $\mathcal{M} = \{max, avg\}$ of number of friends per student and $\mathcal{A}$ being all algorithms; we also report $std$ - the standard deviation for $avgQ$. For all datasets we used the $LinNorm$ project preference function and $\alpha = 10$.**

| Dataset | Algorithm | $maxQ_G$ | $avgQ_G$ | $std$ |
|---------|-----------|----------|----------|-------|
| *Class-A* | Quadratic | 3 | 0.71 | 0.79 |
| | Pipage | 2 | 0.15 | 0.37 |
| | RPipage | 1 | 0.10 | 0.30 |
| | Greedy | 3 | 0.65 | 0.74 |
| | Random | 0 | 0.00 | 0.00 |
| | Manual | 3 | 0.63 | 0.63 |
| *Class-B* | Quadratic | 2 | 0.64 | 0.77 |
| | Pipage | 2 | 0.5 | 0.63 |
| | RPipage | 2 | 0.43 | 0.62 |
| | Greedy | 3 | 0.79 | 1.08 |
| | Random | 2 | 1.5 | 0.87 |
| | Manual | 2 | 0.64 | 0.67 |
| *Class-C* | Quadratic | 1 | 0.54 | 0.50 |
| | Pipage | 1 | 0.38 | 0.49 |
| | RPipage | 1 | 0.46 | 0.50 |
| | Greedy | 2 | 0.69 | 0.67 |
| | Random | 1 | 0.46 | 0.50 |
| | Manual | 2 | 0.69 | 0.54 |
| *Class-D* | Quadratic | 3 | 0.43 | 0.72 |
| | Pipage | 1 | 0.23 | 0.42 |
| | RPipage | 1 | 0.29 | 0.45 |
| | Greedy | 2 | 0.38 | 0.54 |
| | Random | 0 | 0.00 | 0.00 |

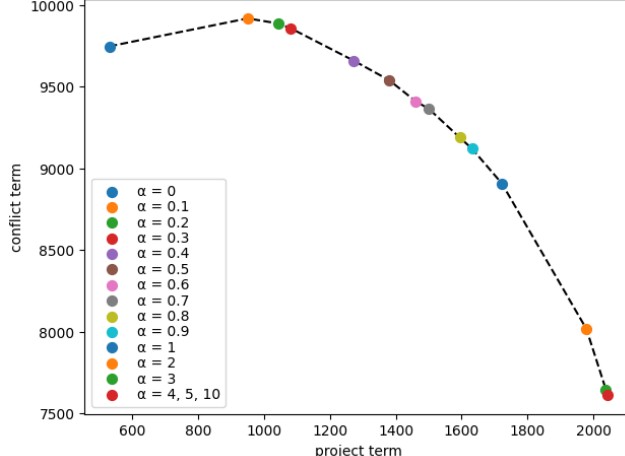

**Figure 12: Employee data; Hyperparameter $\alpha$ tuning for the $Company$ dataset.**

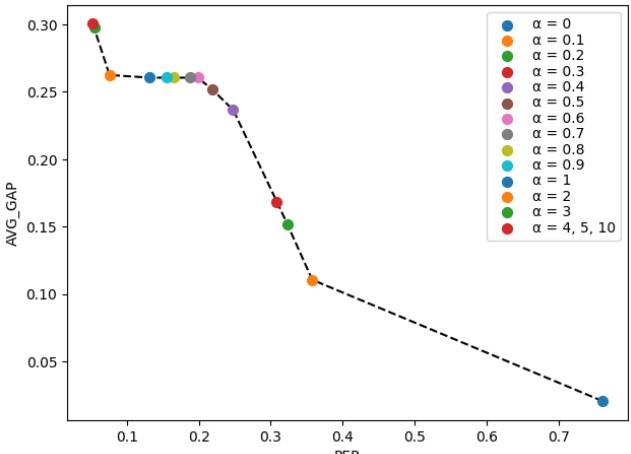

**Figure 13: Employee data; Controlling the balance between the fraction of people who changed department $PER$ and average gender gap per department $AVG\text{-}GAP$ using $\alpha$.**

**Table 6: Education data; $\mathcal{M}Q_R(x_{\mathcal{A}})$ for $\mathcal{M} = max, avg$ of assigned project preferences and $\mathcal{A}$ being all algorithms; we also report $std$ - the standard deviation for $avgQ_R$. For all datasets we used the $LinNorm$ project preference function and $\alpha = 10$.**

| Dataset | Algorithm | $maxQ_R$ | $avgQ_R$ | $std$ |
|---------|-----------|----------|----------|-------|
| *Class-A* | Quadratic | 14.0 | 1.79 | 2.42 |
| | Pipage | 14.0 | 1.81 | 2.47 |
| | RPipage | 14.0 | 1.81 | 2.48 |
| | Greedy | 14.0 | 1.95 | 2.50 |
| | Random | 14.0 | 7.20 | 4.33 |
| | Manual | 14.0 | 2.79 | 2.44 |
| *Class-B* | Quadratic | 4.0 | 1.57 | 0.86 |
| | Pipage | 4.0 | 1.57 | 0.90 |
| | RPipage | 4.0 | 1.57 | 0.90 |
| | Greedy | 7.0 | 2.18 | 1.79 |
| | Random | 7.0 | 4.36 | 2.24 |
| | Manual | 3.0 | 1.71 | 0.80 |
| *Class-C* | Quadratic | 6.0 | 1.62 | 1.11 |
| | Pipage | 6.0 | 1.62 | 1.15 |
| | RPipage | 6.0 | 1.62 | 1.15 |
| | Greedy | 6.0 | 2.12 | 1.48 |
| | Random | 6.0 | 3.54 | 1.67 |
| | Manual | 6.0 | 2.04 | 1.09 |
| *Class-D* | Quadratic | 1.875 | 1.05 | 0.20 |
| | Pipage | 1.875 | 1.05 | 0.20 |
| | RPipage | 1.875 | 1.05 | 0.20 |
| | Greedy | 1.875 | 1.05 | 0.20 |
| | Random | 7.125 | 4.17 | 1.92 |

than the baselines. An observation is that using the $LinNorm$ project preference function does not make a significance difference in the qualitative results. Note again that our main focus on the education datasets was assigning students to projects they like. Assigning them with friends was a secondary goal.

## D.7 Hyperparameter tuning for the *Company* dataset

Figure 13 shows the trade-off between the percentage of people who changed department and the average percentage of the male-female gap per department. Note that for values of $\alpha$ between 0.5 and 2 there is a plateau. That is, the average gender gap per department remains almost constant although more employees change department. After $\alpha$ drops below 0.5 the average male-female gap drops significantly, but the percentage of people who change departments grows very fast. According to the above plot a reasonable choice for $\alpha$ is $\alpha = 2$.

Figure 12 shows the trade-off between the task and social satisfaction terms.

