# OpenReview forum: "Team formation amidst conflicts"
_ACM.org/TheWebConf/2024/Conference — TheWebConf24 Oral_

### Official Review · Reviewer_j8dd · 2023-11-11

**Novelty:** 4
**Technical Quality:** 5

**Review:**

Summary:

The paper proposes and studies the "team formation amidst conflicts" problem, which is a generalization of max-cut. They provide two approximation algorithms and some heuristic ideas to speed-up the running time. They also perform experiments on real-world and synthetic datasets.

Strengths:
1) successfully using, on real-world datasets, dependent-rounding techniques (that the authors claim have not been evaluated experimentally before)
2) even though the novelty of the algorithms is limited (see Weaknesses), the authors provide an equivalent linear formulation which is much more efficient than the concave formulation

Weaknesses:

The novelty of the paper seems limited, in fact:
1) the deterministic 1/2-approximation was, essentially, already known in [1]
2) the rounding scheme of the new 3/4-approximation was already known in [9]; therefore the only novel part of this algorithm seems to be the $L_2$-relaxation
3) it seems that the linear program formulations of the problem were (essentially) known too (equations (21)-(26) of [1]) (of course, except for the new linear term introduced in the objective function)

others:

4) the 3/4-approximation works only for $\lambda$ large enough (while this assumption is not needed in the 1/2-approximation)

Edit: I thank the authors for their clarifications, I increased the novelty score to 4 and the technical score to 5.

**Questions:**

1) for Pipage rounding to work, the objective function must satisfy what [1] calls $\epsilon$-convexity (definition 1 of [1]), is it trivial that this property is respected by your objective function? (it does not seem to be proved or mentioned in the manuscript)
2) In the proof of Proposition 3, the inequality at line 1115 seems to follow from Lemma 2, however, the inequality in Lemma 2 is actually reversed, is this a typo?
3) The heuristic in "sparsification" is said to work well in practice, but it seems to have been tested only on the synthetic dataset. Is this claim based only on the synthetic results?

Other comments:

4) in the computational speed-up section, if I understand correctly, converting the convex programs to linear programs does not alter the theoretical guarantees; while the ideas in "sparsification" and "compact" are heuristics that might decrease the approximation of the algorithms. It would be useful to make these points explicit.
5) Sparsify-Linear and Compact-Linear can solve the synthetic instance quite quickly; it would be useful to try larger synthetic instances to see how much they can scale (e.g., given 1 minute of budget, what is the largest instance solvable by these algorithms?)

Minor:

- at line 409, an expected value is probably missing for the first two terms

**Ethics Review Description:**

There are no ethical issues

**Reviewer Confidence:**

3: The reviewer is confident but not certain that the evaluation is correct

**Scope:**

3: The work is somewhat relevant to the Web and to the track, and is of narrow interest to a sub-community

---

### Official Review · Reviewer_gTjf · 2023-11-21

**Novelty:** 4
**Technical Quality:** 4

**Review:**

This paper addresses the challenge of team formation in conflict-laden scenarios, aiming to assign individuals to tasks while considering their capacities, preferences, and conflicts. The authors introduce efficient approximation algorithms, leveraging dependent rounding schemes as a core tool. Their versatile framework finds applications in various real-world contexts, from education to human resource management. Empirical evaluations on real datasets demonstrate that the proposed algorithms outperform natural baselines, particularly excelling in educational settings where they surpass human experts' manual assignments. In the domain of human resource management, the algorithms also contribute to team diversity. Additionally, experiments with synthetic data confirm the algorithms' scalability in practical applications.

Strengths:
+ A variety of evaluations: 2 real datasets, 1 synthetic data, 4 baselines, quantitative and qualitative studies.
+ Detailed technical proofs and analyses with detailed appendix.
+ Detailed problem definition and an interesting proof showing the team formation amidst conflicts problem is NP-hard.

Weaknesses:
+ Motivation is not clear. It would be great to have some motivating example and some illustration about the current limitations of existing works, the problem, and the solution intuition.
+ Contribution is not clear. The paper proposes some speedup heuristics for approximation but it is really hard to follow.
+ Presentation can be improved greatly. The framework is not clear and it is really difficult to understand the differences between computational speedups and why they work.

Updated: I have read the rebuttal(s).

**Questions:**

+ Motivation is not clear. It would be great to have some motivating example and some illustration about the current limitations of existing works, the problem, and the solution intuition.
+ Contribution is not clear. The paper proposes some speedup heuristics for approximation but it is really hard to follow.
+ Presentation can be improved greatly. The framework is not clear and it is really difficult to understand the differences between computational speedups and why they work.

**Reviewer Confidence:**

3: The reviewer is confident but not certain that the evaluation is correct

**Scope:**

3: The work is somewhat relevant to the Web and to the track, and is of narrow interest to a sub-community

---

### Official Review · Reviewer_BuwB · 2023-11-21

**Novelty:** 4
**Technical Quality:** 5

**Review:**

This paper studied the team formation problem under both preference and conflict constraints. The key idea is to relax the problem as a quadratic optimization problem, and round the non-integer solution to a valid solution via the proposed Relax-round algorithm and its variants. Theoretical analysis on the approximation quality and empirical evaluation on real-world and synthetic datasets are carried out to evaluate the proposed method.

- Pros:
	- The paper is overall well-written and easy to follow.
	- Theoretical results are sounded and help evaluate the proposed approximation algorithm.
	- Variants dealing with sparse and compact data are further proposed for better scalability.

- Cons:
	- More illustration is recommended for the proposed variants (sparse and compact).
	- Hyperparameter study on $\alpha$ can be improved.
	- Visualization can be improved.

**Questions:**

- Line 68: this should be “minimize the sum of conflict edges”?
- Line 335: what is a matching $M_1$?
- Section 5 Compact variant:
	- Can you elaborate how nodes are partitioned into supernodes? What algorithm you use? How do you guarantee that “nodes in A are mostly in conflict with nodes in B” (line 528)?
	- When unrolling the supernode solution to the original graph, how can you guarantee its (sub)optimality? I think it could serve as a good warm-up start point, but can not be directly used as the fractional solution.
	- Does this approach provide a way to perform team formation hierarchically? i.e., you may also have supernode of supernodes?
- Experiments:
	- Table 1: missing bottom line.
	- Selection for $c_{u,t}$: in Education data, $c_{u,t}$ is a fractional value for inverse and linnorm (possibly quite small when rank is high), but in Employee data, it’s a sparse binary value. Since $c_{u,t}$ is crucial for the quality of the solution, can you explain your choices here? Besides, is the proposed method sensitive to different ways of constructing $c_{u,t}$?
	- Hyperparameter study on $\alpha$: in the experiments, you select $\alpha=10$, but when studying the effect of $\alpha$ in Figure 10-13, you did not explore $\alpha$ with values over 10.
- Line 676: Inverse -> LinNorm
- Visualization: several figures (e.g., Fig. 3,5,8,10,11,12,13) are of low-resolution, please include a high-resolution figure in the future version.

**Reviewer Confidence:**

3: The reviewer is confident but not certain that the evaluation is correct

**Scope:**

3: The work is somewhat relevant to the Web and to the track, and is of narrow interest to a sub-community

---

### Official Review · Reviewer_SdBp · 2023-11-23

**Novelty:** 4
**Technical Quality:** 6

**Review:**

This paper presents an algorithmic approach to formulate teams in consideration with conflict information. While generally formulated as Max-k-Cut problem, the team formulation is a practical side of how this can be applied to. The proposed algorithm advances with better efficiency and an additional notion introduced with a linear term added to the problem. The writing across all sections clearly conveys the core idea and all details, starting from Introduction part giving a clear overview of what this study claims, to Evaluation section narrating the results by clearly comparing how algorithms differ in their performance in multiple settings.

Strengths
+ The algorithmic solution has a great impact on many applications
+ Rigor of methodological development and its clear representation
+ Comprehensive evaluations, especially by adding quality validation with a domain expert

Weaknesses
- More discussions on potential applications and problem domains could have been elaborated

**Questions:**

While all details on this study are clearly written, more discussions might be appreciated by readers -- its wider application domains and future work. Although termed as “conflict”, I wonder how this component can be applied, and what other scenarios this algorithm could be used for real-world practices. If there are any future work in consideration with more complex settings that the authors identified, it would be helpful for scholars to take a peek at how they can advance the research beyond this work.

**Reviewer Confidence:**

3: The reviewer is confident but not certain that the evaluation is correct

**Scope:**

4: The work is relevant to the Web and to the track, and is of broad interest to the community

---

### Official Review · Reviewer_3LJL · 2023-12-01

**Novelty:** 4
**Technical Quality:** 5

**Review:**

This study formulates a novel team formation problem considering task preferences and conflicts, employing efficient approximation algorithms based on dependent rounding schemes. Demonstrating versatility, it applies across educational and human-resource management contexts, outperforming natural baselines and manual assignments by experts while enhancing team diversity. Validation through real-world datasets and synthetic tests underscores its scalability and practical effectiveness, paving the way for advanced team optimization algorithms.

**Questions:**

I am not an expert in this area. However, my major concern is that whether your optimization techniques, including the NP-hard proof and the approximation algorithm, is novel enough compared to the typical Min-cut problem and its approximation approaches.

**Ethics Review Description:**

No ethics issue.

**Reviewer Confidence:**

2: The reviewer is willing to defend the evaluation, but it is likely that the reviewer did not understand parts of the paper

**Scope:**

3: The work is somewhat relevant to the Web and to the track, and is of narrow interest to a sub-community

---

### Decision · Program_Chairs · 2024-01-22

**Decision:**

Accept (Oral)

**Comment:**

The paper introduces a reasonably well-motivated generalisation of Max Cut to staudy the problem of team formation among conflicts, providing new algorithms and efficient heuristics that are comprehensively tested with real-world data. From the algorithmic standpoint the contribution is technically interesting for this conference (which is not a pure theory conference) and the good experimental performance of the methods proposed show good applicability to real-world scenarios of a well-motivated problem.